# Synergy Effect and Symmetry-Induced Enhancement Effect of Surface Multi-Defects on Nanohardness by Quasi-Continuum Method

**DOI:** 10.3390/ma15072485

**Published:** 2022-03-28

**Authors:** Zhongli Zhang, Can Wang, Yushan Ni

**Affiliations:** 1Shanghai Institute of Measurement and Testing Technology, Shanghai 201203, China; zhangzl@simt.com.cn (Z.Z.); wangc@simt.com.cn (C.W.); 2Department of Aeronautics and Astronautics, Fudan University, Shanghai 200433, China

**Keywords:** quasicontinuum method, surface multi-defects, synergy effect, symmetry-induced enhancement effect

## Abstract

The quasicontinuum method has been applied to probe the thin film with surface multi-defects, which is commonly seen in nanoimprint technique and bulk micromachining. Three unilaterally distributed multi-defect models and six bilaterally distributed multi-defect models of Pt thin film have been carried out in nanoindentation. The results show that the nanohardness gradually decreases as the number of unilaterally distributed multi-defects increases, along with the increasingly low decline rate of the nanohardness. The synergy effect of the unilaterally distributed multi-defects has been highly evidenced by the critical load revision for dislocation emission of Pt thin film, and it is predicted into a universal form with the synergy coefficient among the existing multi-defects for FCC metals. Moreover, the nanohardness obviously increases when the bilaterally distributed multi-defects form into symmetrical couple, and it could be even greater than the one with defect-free surface, due to the symmetry-induced enhancement effect on nanohardness. The symmetry-induced enhancement coefficient has been brought out and has well explained the symmetry-induced enhancement effect of bilaterally distributed multi-defects on the nanohardness by a prediction formula. Furthermore, the characteristic length of symmetric relations has been brought out to calculate the symmetry-induced enhancement coefficient and it has been effectively predicted to equal to the sum of the adjacent distance between the surface defect and the indenter, the defect depth near the indenter, and the defect width for FCC metal.

## 1. Introduction

It is known that nanoindentation [1] has already been a relatively simple and effective method for evaluating the material property of thin films. Since the nanoindentation on the defect-free surface has been sufficiently investigated, the simulations or experiments on the thin film with defects has become a hot topic recently to get closer to real system such as surface scratches [2], surface steps [3,4] and surface roughness [5,6,7], where the surface roughness is usually treated as the mixed group of surface pit defects [8], which is in fact used to simulate the theoretical numerical investigations, typically and commonly seen in the nanoimprint techniques [9,10,11] and epitaxial thin films [12]. Ni yushan, et al. have simulated the nanoindentation on the pitted surface and have figured out the delay effect [13], the size effect [14], and the distance effect [15] of the surface pit defect. However, relevant researches are only focused on the case of single defect. Moreover, as the MEMS (Micro-Electro-Mechanical System) highly developed into nano-scale, the group of surface pit defects is becoming a common practice in the bulk micromachining [16,17,18,19], especially in the dry etching field [20,21]. So it is in great significance to simulate the thin film with surface multi-defects; however, relevant investigations on this aspect are still rare to see. Our aim is to probe the mechanism how the surface multi-defects influence the material property in nanoindentation by Quasicontinuum method (QC), which is a multiscale method that applies molecular dynamics (MD) model at the intense deformation region and finite element model elsewhere so that it actually has great advantages in saving calculating time with PC environment under accuracy ensurance than the one all through MD method.

## 2. Methodology

Quasicontinuum method [22] with Ercolessi-Adams potential (EAM) [23] is applied in this simulation, which proceeds through molecular static energy minimization over an atomistic (non-local) domain to describe the atomistic behavior and a finite element (local) domain at the linear deformation area of the material.

The schematic representation of the nanoindentation model with unilaterally and bilaterally distributed multi-defects with local and non-local representative atoms in the simulation are shown in Figure 1 and Figure 2. The orientation is selected to facilitate dislocation emission, where the *x*-axis direction is [1 1 1], the *y*-axis direction is $\left[\bar{1}10\right]$ and the outer-of-plane z direction is $\left[\bar{1}\bar{1}2\right]$. The material of model is single crystal Pt thin film, and as the crystallographic lattice constant (a_0_) of Pt is 0.392 nm. When it is projected onto the XY plane, the adjacent distance between atoms in [1 1 1] direction (d_0_) is 0.2263 nm while in $\left[\bar{1}10\right]$ direction (h_0_) it is 0.1386 nm. The length of the Burgers vector ($\vec{b}$) is 0.277 nm. Poisson *ν* is 0.39 and (1 1 1) surface energy γ_111_ is 1.47 J/m^2^ [24]. The value of the shear modulus μ can be calculated through the parameters in EAM potential by the equation $\mu =\frac{1}{5}\left(C_{11}-C_{12}+3C_{44}\right)$ [25], which gives μ = 66 GPa of Pt crystal, where ***C***_11_ = 302 GPa, ***C***_12_ = 206 GPa and ***C***_44_ = 78 GPa.

The rigid indenter with the width of 8d_0_, is driven into the Pt thin film in its size of 200 nm width and 100 nm height, which ensures that far-field boundary conditions do not affect the behavior in the vicinity of indenter. Besides, the thickness of this model is 0.4801 nm, equal to the minimal repeat distance with periodic boundary condition applied. The adjacent distance among the surface pit defects is selected to be 3d_0_ (as shown in Figure 1 and Figure 2), in order to avoid the distance influence caused by the periodic arrangement “ABCABC” of the atoms in face-centered cubic metal based on previous research [15]. Three unilaterally distributed multi-defects models, N1, N2 and N3, have been carried out in this simulation shown as Figure 1, respectively one, two and three surface pit defects with the fixed size of 3d_0_ width and 5h_0_ height. Furthermore, six bilaterally distributed multi-defects models, R1, R1−L1, R2−L1, R2−L2, R3−L2 and R3−L3, have been carried out in order to make a more comprehensive investigation, shown as Figure 2, where the surface pit defect with the fixed width 3d_0_ (W), and increases every 3h_0_ (H) in the height at the both side of indenter in turn.

Moreover, the boundary condition of the Pt thin film in the out-of-plane direction keeps rigid at bottom and is free at the sides. The atoms under the indenter are forced to move gradually into the material by displacement-imposed boundary conditions. Each load step of the indenter is 0.02 nm, with a final depth 1.0 nm, which is highly effective and efficient to catch the details of the elastic-to-plastic transition. Only 5000 atoms are treated explicitly at most (15,000 degrees of freedom), and one single simulation can be finished on a common personal computer in five days. It is necessary to notice that our aim is to investigate how the distribution of the surface multi-defects influences the nanohardness and its micro mechanism. Thus, it is a great difficulty to make a Pt thin film sample with specific surface multi-defects. Any difference between those samples will influence the results. In fact, a QC method where the conditions can be precisely and singly controlled is conductive to such an aim, and this has already been verified as undeniable, valid and reliable for decades through theoretical analysis and experiments [26,27,28,29].

## 3. Results and Discussion

### 3.1. Unilaterally Distributed Multi-Defects

In the present paper, three unilaterally distributed multi-defect models, marked as N1, N2 and N3, have been carried out in this simulation in order to probe the synergy effect of unilaterally distributed multi-defects on nanohardness. Figure 3 shows the load-displacement curve of the Pt thin film of the simulation model of unilaterally distributed multi-defects, where the load is expressed as a force per unit thickness (N/m), which is obtained by dividing the total force on the indenter by the repeat distance in the out-of-plane direction of the crystal structure (0.4801 nm for this model). The critical load for elastic-to-plastic transition is corresponding to the first peak point at the linear section of the load-displacement curve, indicating the onset of plasticity of the Pt thin film.

It can be easily found that the yield load of the Pt thin film can be obviously influenced by the surface pit defect. Furthermore, the yield load of the Pt thin film decreases in general and the load at the stable dislocation emission gradually decreases from 26.85 N/m to 23.78 N/m as the surface pit defect increases results in an increasingly lower energy of system. In order to make a comprehensive investigation of unilaterally distributed multi-defects, the nanoindentation on a free surface has been carried out for comparison, where the Pt thin film turns from elasticity to plasticity in only one step. However, the nanoindentation with the surface pit defects all experience fluctuant sections in the load–displacement curve, which means there is a dislocation reaction [13].

The nanohardness curve of the simulation model of unilaterally distributed multi-defects has been plotted in Figure 4 to show how the surface pit defects distributed in the unilateral side of the indenter influences the nanohardness. It can be easily found that the nanohardness (the maximum load in Figure 3 divided by the width of the indenter) of the Pt-thin film gradually decreases from 16.87 GPa to 16.03 GPa as the number of defects increases. Furthermore, the decline rate of the nanohardness is getting slower. According to the relevant nanoindentation experiment, the hardness of Pt films are in the range of 3.3–7.5 GPa, after removing the residual stress in the range of 839–1720 MPa by annealing [30]. The nanohardness of the Pt-thin film in this simulation is approximately as large as the experiment result at the magnitude.

Moreover, the snapshot of atoms under the indenter and in the corresponding out-of-plane displacement plot in the simulation model of unilaterally distributed multi-defects before and after the dislocation emission have been probed as shown in Figure 5, where the dislocation nucleated beneath the indenter at the load peak of the load-displacement curve, and was finally emitted in the stable way of two Shockley partial dislocations deep in the Pt thin film.

Based on the emission depth of the dislocation obtained in Figure 5, there is a great need to further discuss the potential mechanism of the unilaterally distributed multi-defect influence of nanohardness according to the calculation formula of the necessary load for the elastic-to-plastic transition that was applied by Tadmor [25] as follows:(1)$$P_{cr}=\frac{\mu b}{4\pi (1-v)}\ln\frac{32h(h+2a)a^{2}}{b^{4}}+2\gamma_{111}+\frac{1}{2}kb$$
where $P_{cr}$ is the critical load value for elastic-to-plastic transition, μ is the shear modulus of Pt crystal, *k* is the slope of elastic stage in the load-displacement curve, *h* is the depth of emitted dislocation dipole, *a* is the half width of indenter, $\gamma_{111}$ is the energy of (111) surface of Pt crystal.

Since Equation (1) is only suitable for the nanoindentation on the free surface, it is necessary to further figure out the relationship between the correction value and the unilaterally distributed multi-defects to meet the critical load value for the elastic-to-plastic transition of Pt from QC simulation.

Table 1 shows the differential value between the *P*_*cr*_ from Equation (1) after environment noise correction and the QC simulation results among the cases of unilaterally distributed multi-defects, where $\Delta_{1}$, $\Delta_{2}$, $\Delta_{3}$ are the corresponding correction values of one, two, three surface pit defects in the simulation model of unilaterally distributed multi-defects, and the ${\Delta^{\prime }}_{1}$, ${\Delta^{\prime }}_{2}$, ${\Delta^{\prime }}_{3}$ are the corresponding correction values of 1st, 2nd, 3rd surface pit defects, as shown in the table. Several repeat simulations have been carried out to verify whether the randomness is the major factor, by which the accuracy of 0.01 N/m for the different delta values in Table 1 has been obtained. It can be found out from the data that:(2)$$\Delta_{2}={\Delta^{\prime }}_{1}+{\Delta^{\prime }}_{2}+1.97\times \left|{\Delta^{\prime }}_{1}\cdot {\Delta^{\prime }}_{2}\right|$$
(3)$$\Delta_{3}={\Delta^{\prime }}_{1}+{\Delta^{\prime }}_{2}+{\Delta^{\prime }}_{3}+2.14\times \left|{\Delta^{\prime }}_{1}\cdot {\Delta^{\prime }}_{2}\cdot {\Delta^{\prime }}_{3}\right|$$

It seems it might be even predicted to the relation that:(4)$$\Delta_{N}=\sum_{i=1}^{N}{\Delta^{\prime }}_{i}+\gamma \times \prod_{i=1}^{N}\left|{\Delta^{\prime }}_{i}\right|$$
where γ is the synergy coefficient among the existing unilaterally distributed multi-defects, which is approximately 2 in this simulation. It is necessary to declare that Equation (4) is just made based on the simulation model details, that is to way, though it is necessary to further probe whether it is still valid if the sizes, adjacent distance or the crystal orientation of the unilaterally distributed multi-defects changes, it is in any event highly evidenced to predict the validity of its general form of synergy effect among the FCC metals. Fortunately, it might be instructive and significant to MEMS or nano-material surface application.

### 3.2. Bilaterally Distributed Multi-Defects

Based on the study of the model of unilaterally distributed multi-defects, the model of bilaterally distributed multi-defects, respectively R1, R1−L1, R2−L1, R2−L2, R3−L2 and R3−L3, have been further carried out according to the simulation model details as shown in Figure 2.

The nanohardness curve of the simulation model of bilaterally distributed multi-defects has been displayed in Figure 6, where the nanohardness expresses a fluctuant curve alternately up and down. The result obviously shows that when the surface pit defects are symmetrically distributed on both sides of the indenter, the nanohardness of the Pt thin film gets a relatively greater value (II, IV, VI sections in Figure 6) than the ones in the unsymmetrical cases (I, III, V sections in Figure 6). That is to say, there is a certain symmetry-induced enhancement effect of multi-defects on nanohardness. Furthermore, the nanohardness of the Pt thin film with symmetrical distributed multi-defects could be even greater than the thin film with no defects due to such an enhancement effect (approximately 2.6% increase according to R2−L2 in Figure 6).

In order to probe the reason for such an obvious decline in nanohardness (V sections in Figure 6), the Von Mises strain distribution of notch propagation at bilaterally distributed multi-defects have been investigated and are shown in Figure 7. This shows that in the simulation models of R3−L2 and R3−L3, notches occurred at the tip of the defects near to the indenter while others had no notches, which is most likely due to the great depth of the surface pit defects with the high concentration strain. It can obviously be figured out according to the comparison of the simulation models of R3−L2 and R3−L3 that, even if there is notch propagation that weakens the nanohardness of the Pt thin film, the symmetry-induced enhancement effect of bilaterally distributed multi-defects on the nanohardness has still played a significant role as strong evidence.

The dislocated structure beneath the rigid indenter is given in Figure 8 along with the out-of-plane displacements experienced by the atoms. Two dissociated <110> edge dislocations have been emitted beneath the indenter tips in the simulation model of “R1”,”R1−L1”, ”R2−L1”, ”R2−L2”, and beneath the defects tips in the simulation model of ”R3−L2” and “R3−L3”. The full dislocation with the Burgers vector has dissociated into two Shockley partials separated by stacking faults.

It is quite possible to explain such a symmetry-induced enhancement effect of bilaterally distributed multi-defects by the grain boundary strengthening effect, which is one of the main strengthening effects in the dislocation theory [31]. Moreover, some literature shows that the symmetric and asymmetric grain boundaries play a great role on the mechanical behavior of materials [32].

As it is known that the lattice resistance would decline to some extent due to the existence of the surface defects, the new grain boundary of the surface defects would induce a certain field of stress because of the boundary effect. Figure 9 shows the schematic illustrations of the edge dislocations with Burgers vector $\vec{b}=\frac{1}{2}[\bar{1}10]$ on Pt (111) plane and the simple model of bilaterally distributed multi-defects, where the red arrows represent the effect of the stress field formed by the grain boundary on the movement of the dislocations.

In order to further explain the symmetry-induced enhancement effect of bilaterally distributed multi-defects on the nanohardness, the symmetry-induced enhancement coefficient (γ) has been defined and the formula of such an enhancement effect on nanohardness can be expressed as follows:(5)$$\Delta_{N}=\gamma \cdot \sum_{i}^{n}\Delta_{\mathrm{BE}}^{i}-\sum_{i}^{n}\Delta_{\mathrm{DF}}^{i}$$
where γ belongs to [0,1], determined by the symmetric relation of the bilaterally distributed multi-defects, namely, it equals 1 if the bilaterally distributed multi-defects are formed into a totally symmetric couple, while it equals 0 if the defects are unilaterally distributed only at one side.

Such an assumption is in great keeping with the above results that the enhancement effect on the nanohardness only exists under circumstance of bilaterally distributed multi-defects. For unilaterally distributed multi-defects, the nanohardness gradually decreases with the increase in the defects; $\Delta_{\mathrm{BE}}^{i}$ is the nanohardness increment induced by the grain boundary stress filed at the *i*th surface defect, which is mainly related to the defect size, the distance between the dislocation slip fields, and the crystal direction of the defect boundary; $\Delta_{\mathrm{DF}}^{i}$ is the nanohardness decrement due to the existence of the ith initial surface defect, which is closely related to the size of the initial surface defect, the distance between the indenter and the defects according to the previous researches [14,15]. It is necessary to explain that Equation (5) has been brought out particularly to analyze the symmetry-induced enhancement effect on the nanohardness in the case of one single surface defect on each side, since the synergy effect of the unilaterally distributed multi-defects has been discussed.

Based on the above discussion, in the simulation model of “R1”, the symmetry-induced enhancement coefficient (γ) is 0, and the influence on the nanohardness of the Pt thin film compared to the defect-free case is as follows, where H and W represent the depth and width of surface defects in Figure 2.
(6)$$\Delta_{N}^{R1}=-\Delta_{\mathrm{DF}}^{H-W}=-0.71\mathrm{GPa}$$

In the simulation model of “R1−L1”, the symmetry-induced enhancement coefficient (γ) is 1 due to the perfectly symmetrically distributed initial surface defects on both sides of the indenter and the influence on the nanohardness of the Pt thin film compared to the defect-free case is as follows:(7)$$\Delta_{N}^{R1-L1}=2\Delta_{\mathrm{BE}}^{H-W}-2\Delta_{\mathrm{DF}}^{H-W}≅0$$

In the simulation model of “R2−L1”, the initial surface defects asymmetrically distributed on both sides of the indenter have a different depth, assuming that the symmetry-induced enhancement coefficient (γ) is $\gamma_{\mathrm{HW}-2\mathrm{HW}}$, and the influence on the nanohardness of the Pt thin film compared to the defect-free case is as follows:(8)$$\Delta_{N}^{R2-L1}=\gamma_{\mathrm{HW}-2\mathrm{HW}}(\Delta_{\mathrm{BE}}^{H-W}+\Delta_{\mathrm{BE}}^{2H-W})-\Delta_{\mathrm{DF}}^{H-W}-\Delta_{\mathrm{DF}}^{2H-W}=-0.17\mathrm{GPa}$$

In the simulation model of “R2−L2”, the symmetry-induced enhancement coefficient (γ) is 1, due to the perfectly symmetrically distributed initial surface defects on both sides of the indenter. The influence on the nanohardness of the Pt thin film compared to the defect-free case is as follows:(9)$$\Delta_{N}^{R2-L2}=2\Delta_{\mathrm{BE}}^{2H-W}-2\Delta_{\mathrm{DF}}^{2H-W}=0.43\mathrm{GPa}$$

The simulation model of single surface defect with 2H depth and W width, has been carried out in order to obtain the influence on the nanohardness of the Pt thin film compared to the defect-free case is as follows where the symmetry-induced enhancement coefficient (γ) is 0.
(10)$$\Delta_{N}^{R2}=-\Delta_{\mathrm{DF}}^{2H-W}=-1.29\mathrm{GPa}$$

To promote the conclusion to be more extensive, the nanohardness increment $\Delta_{\mathrm{BE}}^{i}$ induced by the grain boundary stress filed from the bilaterally distributed multi-defects has been divided by the shear modulus μ of Pt.

Then, combined with Equations (6)–(10), the influence of the stress field from the bilaterally distributed multi-defects on nanohardness and the corresponding symmetry-induced enhancement coefficient can be calculated as follows:(11)$$\left\{ \begin{array}{l} \gamma_{\mathrm{HW}-2\mathrm{HW}}=0.826\\ \Delta_{\mathrm{BE}}^{H-W}=1\times 10^{-2}\cdot \mu \\ \Delta_{\mathrm{BE}}^{2H-W}=2.3\times 10^{-2}\cdot \mu \end{array}\right.$$

It can be easily seen that the symmetry-induced enhancement coefficient $\gamma_{\mathrm{HW}-2\mathrm{HW}}=0.826$, exactly fall in the range of its definition between 0 and 1, suggesting that the prediction model (Equation (5)) for the symmetry-induced enhancement effect on nanohardness has a certain reasonability. Moreover, it can be found that the nanohardness increment induced by the grain boundary stress field of the surface defect with 2H depth and W width is about 2 times as large as the one of the surface defect with H depth and W width, which means that the nanohardness increment induced by the grain boundary stress field is mainly related to the depth of the surface defect.

It is significant to have a further discussion about how the symmetry-induced enhancement coefficient is determined. Based on the simulation model of “R2−L1”, the characteristic length of the symmetric relation has been carried out to calculate the symmetry-induced enhancement coefficient. Table 2 shows the calculation of the symmetry-induced enhancement coefficient and the corresponding characteristic length of the symmetric relation in the simulation model of “R2−L1”.

The first column of Table 2 shows the three cases of the characteristic length of symmetric relations, where the characteristic length has been marked as red. Based on the parameters of the simulation model and the lattice parameter of Pt, the value of the symmetry-induced enhancement coefficient can been calculated according to the following formula.
(12)$$\gamma =\frac{\min(\sum_{i}^{n}L_{i}|\mathrm{at}\;\mathrm{one}\;\mathrm{side},\sum_{j}^{m}L_{i}|\mathrm{at}\;\mathrm{the}\;\mathrm{other}\;\mathrm{side})}{\max(\sum_{i}^{n}L_{i}|\mathrm{at}\;\mathrm{one}\;\mathrm{side},\sum_{j}^{m}L_{i}|\mathrm{at}\;\mathrm{the}\;\mathrm{other}\;\mathrm{side})}$$
where the symmetry-induced enhancement coefficient equals the one that the sum of the characteristic length on one side of the indenter with the smaller value divided by the sum of the characteristic length on the other side of the indenter with the larger value. According to the second case in Table 2, the calculation result of the symmetry-induced enhancement coefficient is 0.811, which is closest to 0.826 from Formula (12). That is to say, it might be reasonable to predict for FCC metal that such a characteristic length is equal to the sum of the adjacent distance between the surface defect and the indenter, the defect depth near the indenter, and the defect width.

## 4. Conclusions

The Pt thin film with surface multi-defects of unilateral and bilateral distribution has been investigated by a Quasicontinuum method in nanoindentation. The major conclusions that can be drawn are as follows:

The nanohardness gradually decreases as the number of unilaterally distributed multi-defects increases, along with the increasingly low decline rate of the nanohardness.The synergy effect of the unilaterally distributed multi-defects on the nanohardness has been highly evidenced by the critical load revision for dislocation emission of the Pt thin film, and it is predicted into a universal form with the synergy coefficient among the existing multi-defects for FCC metals.The nanohardness obviously increases along with the increasing emission depth of dislocation at the micro level when the bilaterally distributed multi-defects form into a symmetrical couple, and it could be even greater than the one with the defect-free surface, due to the symmetry-induced enhancement effect on nanohardness.The symmetry-induced enhancement coefficient γ has been brought out and has well explained the symmetry-induced enhancement effect of bilaterally distributed multi-defects on the nanohardness by a prediction formula.The characteristic length of the symmetric relation has been brought forward to calculate the symmetry-induced enhancement coefficient and it has been effectively predicted to equal the sum of the adjacent distance between the surface defect and the indenter, the defect depth near the indenter, and the defect width for the FCC metal.

## Figures and Tables

**Figure 1 materials-15-02485-f001:**
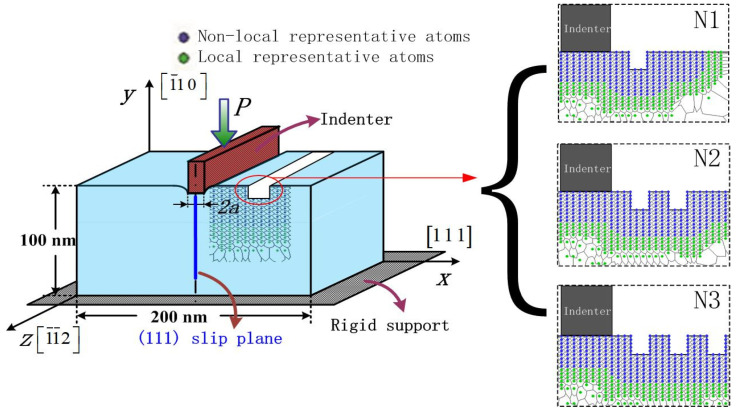
Schematic representation of the nanoindentation models of unilaterally distributed multi-defects: (N1) one surface pit defect of 3d_0_ width and 5h_0_ height; (N2) two surface pit defects of 3d_0_ width and 5h_0_ height; (N3) three surface pit defects of 3d_0_ width and 5h_0_ height.

**Figure 2 materials-15-02485-f002:**
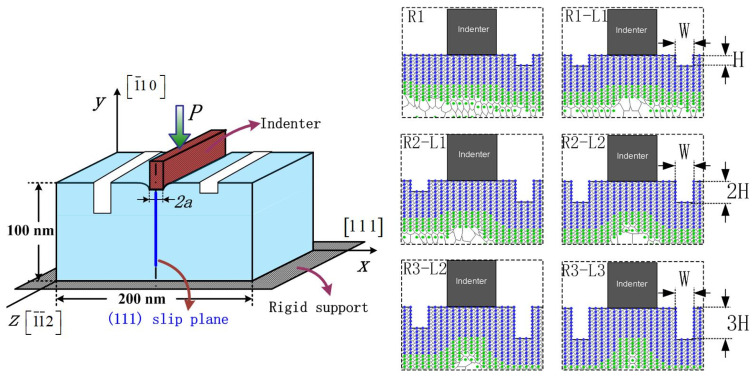
Schematic of local and non-local representative atoms of bilaterally distributed multi-defects. (R1) one pit defect of 3d_0_ width (W) and 3h_0_ height (H) at the right side of indenter; (R1−L1) two pit defects of W width and H height at the both side of indenter; (R2−L1) one pit defect of W width and 2H height at the right side and the other pit of W width and H height at the left; (R2−L2) two pit defects of W width and 2H height at the both side of indenter; (R3−L2) one pit defect of W width and 3H height at the right side and the other pit of W width and 2H height at the left; (R3−L3) two pit defects of W width and 3H height at the both side of indenter.

**Figure 3 materials-15-02485-f003:**
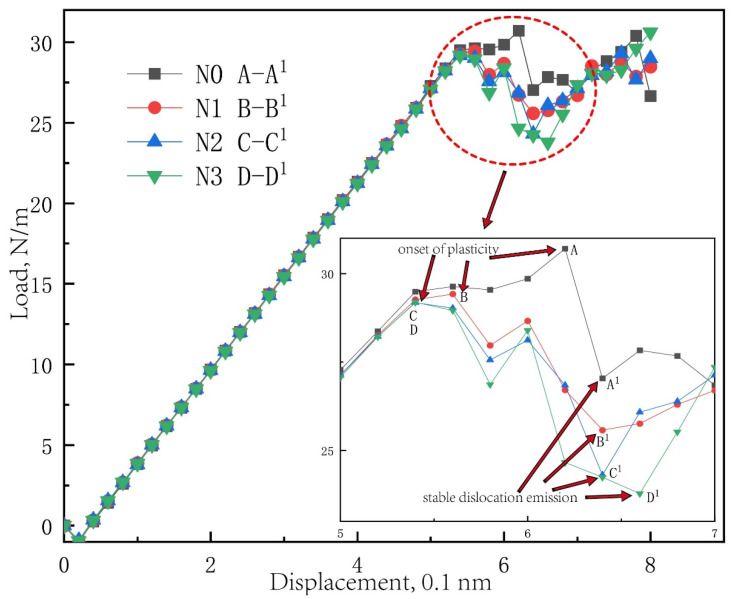
The load-displacement curve of Pt thin film of the simulation model of unilaterally distributed multi-defects: (N0 A–A^1^) the maximum load A and minimum load A^1^ of defect-free model; (N1 B–B^1^) the maximum load B and minimum load B^1^ of N1 model; (N2 C–C^1^) the maximum load C and minimum load C^1^ of N2 model; (N3 D–D^1^) the maximum load D and minimum load D^1^ of N3 model.

**Figure 4 materials-15-02485-f004:**
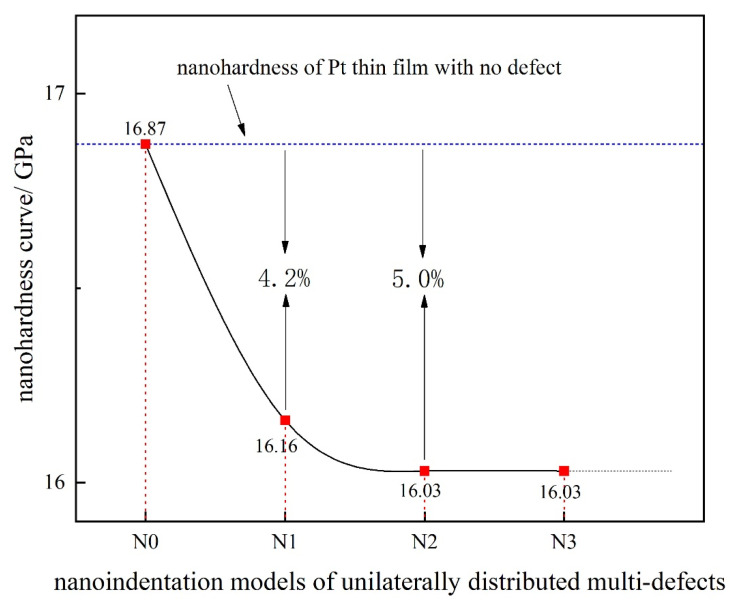
The nanohardness curve of the simulation model of unilaterally distributed multi-defects.

**Figure 5 materials-15-02485-f005:**
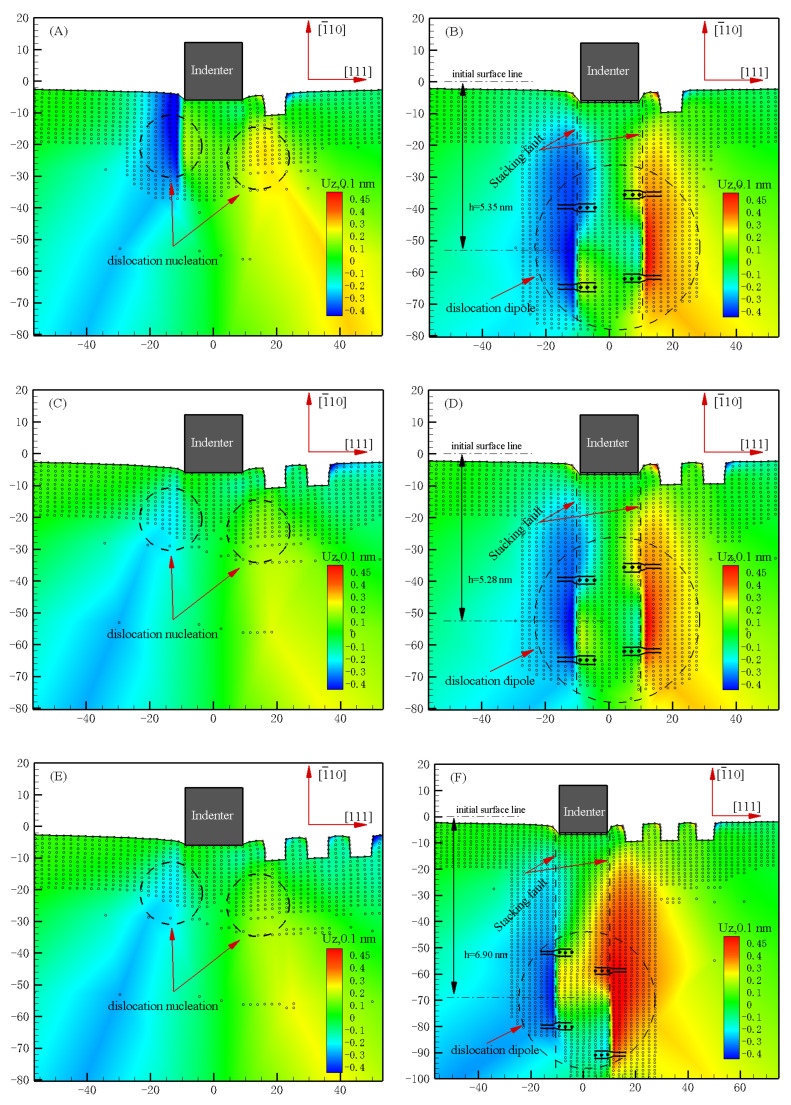
Snapshot of atoms under indenter and corresponding out-of-plane displacement plot in the simulation model of unilaterally distributed multi-defects, where the UZ is the atom displacement at out-of-plane: (**A**) N1 B in Figure 3 (dislocation nucleation); (**B**) N1 B^1^ in Figure 3 (stable dislocation emission at the indentation depth of 0.64 nm); (**C**) N2 C in Figure 3 (dislocation nucleation); (**D**) N2 C^1^ in Figure 3 (stable dislocation emission at the indentation depth of 0.64 nm); (**E**) N3 D in Figure 3 (dislocation nucleation); (**F**) N3 D^1^ in Figure 3 (stable dislocation emission at the indentation depth of 0.66 nm).

**Figure 6 materials-15-02485-f006:**
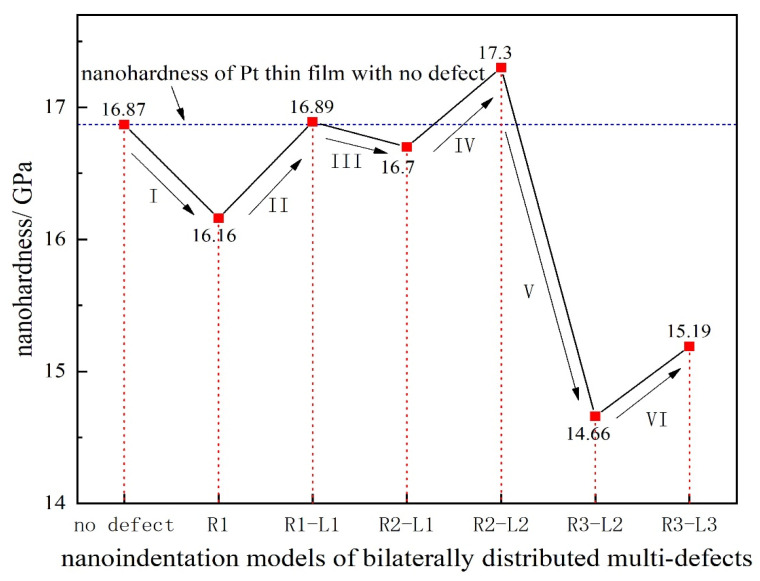
The nanohardness curve of the simulation model of bilaterally distributed multi-defects.

**Figure 7 materials-15-02485-f007:**
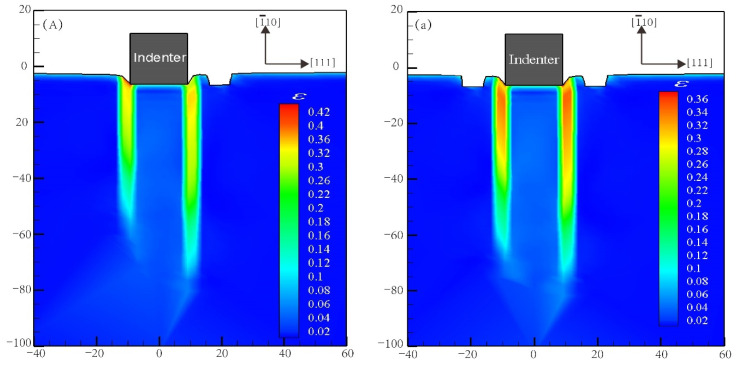
Von Mises strain distribution of notch propagation at bilaterally distributed multi-defects. (**A**) the red point of “R1” in Figure 6; (**a**) the red point of “R1−L1” in Figure 6; (**B**) the red point of “R2−L1” in Figure 6; (**b**) the red point of “R2−L2” in Figure 6; (**C**) the red point of “R3−L2” in Figure 6; (**c**) the red point of “R3−L3” in Figure 6.

**Figure 8 materials-15-02485-f008:**
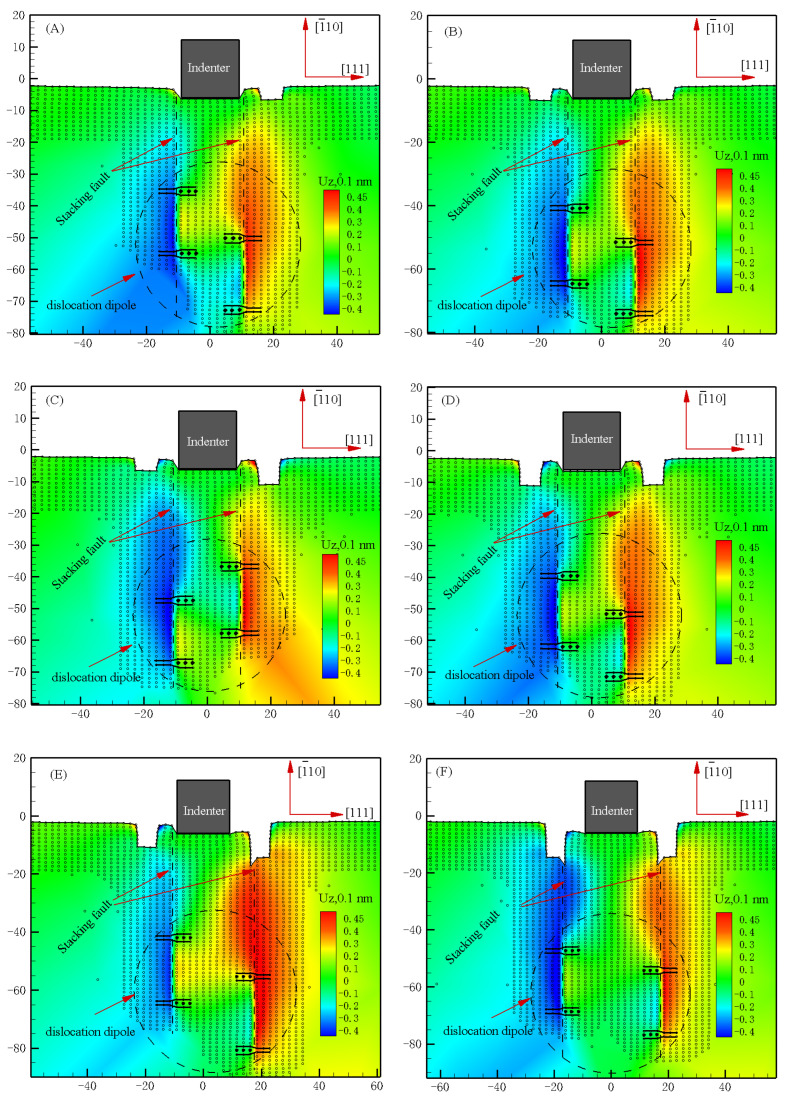
Snapshot of atoms under the indenter and corresponding out-of-plane displacement plot in the simulation model of the bilaterally distributed multi-defects, where UZ is the atom displacement at out-of-plane (dimensions in 0.1 nm): (**A**) the red point of “R1” in Figure 6 (at the indentation depth of 0.62 nm); (**B**) the red point of “R1−L1” in Figure 6 (at the indentation depth of 0.62 nm); (**C**) the red point of “R2−L1” in Figure 6 (at the indentation depth of 0.62 nm); (**D**) the red point of “R2−L2” in Figure 6 (at the indentation depth of 0.64 nm); (**E**) the red point of “R3−L2” in Figure 6 (at the indentation depth of 0.60 nm); (**F**) the red point of “R3−L3” in Figure 6 (at the indentation depth of 0.60 nm).

**Figure 9 materials-15-02485-f009:**
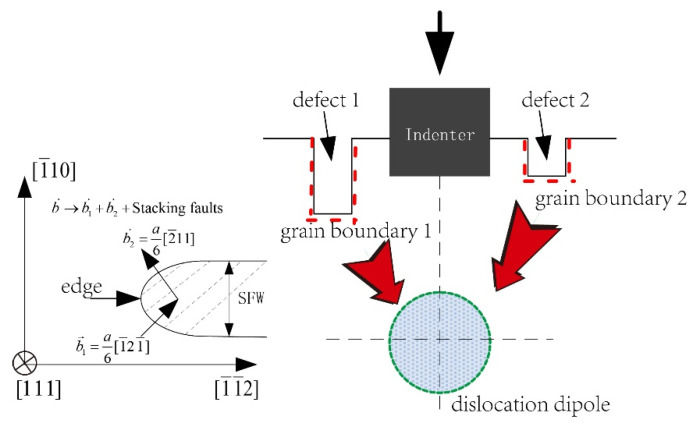
The Schematic illustrations of edge dislocations with Burgers vector $\vec{b}=\frac{1}{2}[\bar{1}10]$ on Pt (111) plane and the model of bilaterally distributed multi-defects, where “SFW” is the stacking fault width.

**Table 1 materials-15-02485-t001:** Correction value in each case of unilaterally distributed multi-defects.

Cases of Unilaterally Distributed Multi-Defects	$P_{cr}$ from Equation (1) after Environment Noise Correction N/m	$P_{cr}$ from QC Simulation as a Standard N/m	Correction Value Δ N/m
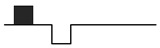 N1	29.81	29.42 ± 0.01	${\Delta^{\prime }}_{1}=\Delta_{1}=-0.39$
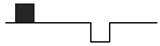 N2-single	30.19	29.41 ± 0.01	${\Delta^{\prime }}_{2}=-0.78$
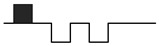 N2	29.75	29.18 ± 0.01	$\Delta_{2}=-0.57$
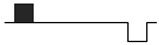 N3-single	30.50	29.64 ± 0.01	${\Delta^{\prime }}_{3}=-0.86$
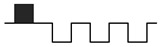 N3	30.65	29.18 ± 0.01	$\Delta_{3}=-1.47$

**Table 2 materials-15-02485-t002:** The characteristic length of the surface defect and corresponding symmetry-induced enhancement coefficient.

The Sketch Map of the Characteristic Length of Symmetric Relation	The Characteristic Length of the Left Surface Defect	The Characteristic Length of the Right Surface Defect	The Symmetry-Induced Enhancement Coefficient γ
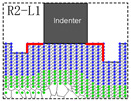	3d_0_ + H	3d_0_ + 2H	0.725
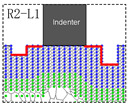	3d_0_ + H + W	3d_0_ + 2H + W	0.811
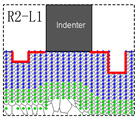	3d_0_ + 2H + W	3d_0_ + 4H + W	0.725

## Data Availability

The research data required to reproduce this work reported in this manuscript is free to obtain by email.

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
