# Peer review of "Synergy Effect and Symmetry-Induced Enhancement Effect of Surface Multi-Defects on Nanohardness by Quasi-Continuum Method"

_materials, 2022, doi:10.3390/ma15072485_

Round 1
Reviewer 1 Report
The manuscript reports quasi-continuum simulation of nano-indentation in a selected geometry of fcc platinum, focussing on the effect of irregularities in the surface (multi-defects) on the simulated load-displacement curves. Such research is definitely relevant for the understanding of nanoindentation of real crystals. I have, however, some considerable concerns with the present version of the manuscript and also with some aspects of the research design:
1. The manuscript is quite incoherent. It appears to be the result of two students’ works put together without too much ado (sections 3.1 and 3.2) and also without big interaction between the chapters. Why is there, e.g. different design of Figure 5 and 8? There does not seem to be a table corresponding to Table 1 in section 3.2. There should be as much as possible parallelism in the presentation of section 3.1 and 3.2.
2. The English is rather poor. This does not only concern some incorrect sentences, but there is also some awkward use of technical terms.
3. It is sometimes difficult to follow the way how some quantities are extracted from the simulations. In particular, while I believe that the effect of the irregularities introduced into the surface on the load-displacement curve does exist, but in my feeling the effects are not very strong. This leads to the question, in which way “randomness” in the simulations might influence the results. This concerns especially the delta parameters complied e.g. in Table 1 and used throughout the paper to quantify the effects of the different types of surface defects. I was unable to figure out how these values have been determined from the load-displacement curves. One problem seems to be the discrete increase of the displacement for the simulations and, of course, the discreteness of dislocation generation. The up and down of the load values in Figure 3 really leads to doubts whether the corresponding results are really robust against slight variations of the simulation conditions. In Table 1 the authors prose an accuracy of 0.01 N/m for the different delta values Table 1. What is the basis for this accuracy? How is P_cr really determined from the QC simulations and how sensitive are these in view of slight variations of conditions? What happens, if the stepsize for displacement is made smaller by a factor of 2?
4. The authors apparently do not discuss details of the orientation of the dislocation dipoles. This orientation is, for example, flipped if you compare Fig. 5 a and b vs. Fig 5c. Why is this? There is asymmetry in the system, so it cannot be accidental. The same asymmetry seems relevant also for the simulations from 3.2. In general, I have the feeling that there no symmetry induced enhancement, but asymmetry induced decrease of hardness!
5. I am completely lost by the discussion on page 17-18. Apparently little reference is made to section 3.1 in that later section. Two independent works put together….
More detailed comments:
(a) Why did the authors study platinum? I note that platinum has a very large stacking fault energy. Is the very high value of that SFE reflected by the employed potential?
(b) Line 52: EAM means embedded atom method. Apparently there are some works by Ercolessi and Adams to determine EAM parameters, but Ref. [19] is not such a work. The Ercolessi and Adams works have appeared later. Please check!
(c) E.g. line 62. The notion “atomic spacing” is inaccurate. The smallest distance between Pt atoms is 0.277 nm, so the “atomic spacing” cannot be smaller.
(d) Line 63 is not be Burgers vector but the LENGTH of the Burgers vector.
(e) Fig. 2 The quantities H and W are not accurately referred to/defined in the Figure caption (Figure + corresponding Figure captions should be selfcontained).
(f) Self contained Figure+Figure caption: A-A1 etc. are unclear from Figure 3.
(g) Figure 4: How is the hardness calculated? The y-axis is unlikely the “nanohardeness curve”
(h) The labelling of the subfigures in Figure 5 is unclever (the same holds for Fig. 7 and 8): simply use the codes of your microstructures N1, L3R3 etc. alongside with possible points (B, B1, holds for Figure 5) directly in the subfigure or use that as the label. The can make the figure captions much more efficient and all is easier for the reader.
(i) Line 168 and 170: there are awkward commas between the delta parameters.
(j) Eq. 8-10: the unit GPa should not be italic.
(k) Figure 8: In my opinion there should be a stacking fault only between the two pair Shockley dislocations constituting the dipole.
(l) Figure 9 is a riddle to me. What is “SFW”?
Author Response
Author response
Dear Reviewer 1,
Thank you for your kind suggestions. All comments have been replied as follows.
1) The manuscript is quite incoherent. It appears to be the result of two students’ works put together without too much ado (sections 3.1 and 3.2) and also without big interaction between the chapters. Why is there, e.g. different design of Figure 5 and 8? There does not seem to be a table corresponding to Table 1 in section 3.2. There should be as much as possible parallelism in the presentation of section 3.1 and 3.2.
- Thanks and we are sorry for such misunderstanding. This work is indeed not the result of two students’ works directly put together, but actually two periods of our research, in which we focus on the distribution effect of surface multi-defects in nanoindentation. That’s why we finally decide to put the unilaterally distributed multi-defects model and the bilaterally distributed multi-defects model together though it might seems to be two separate individuals but indispensible complementary parts, in order to make a comprehensively investigation of distribution effect of surface multi-defects. According to the suggestion, Figure 5 and 8, as well as the two simulation models in Figure 1, have been re-drew to be the same design.
Secondly, we found two different effects respectively shows in the simulation modes, where the synergy effect in the unilaterally distributed multi-defects model (section 3.1) and the symmetry-induced enhancement effect in the bilaterally distributed multi-defects mode (section 3.2). So, the Table 1 in section 3.1 is only suitable for the synergy effect.
2) The English is rather poor. This does not only concern some incorrect sentences, but there is also some awkward use of technical terms.
- Thank you for your suggestion. We are sorry for that. Some incorrect sentences and technical terms have been modified as well as possible through the whole manuscript.
3) It is sometimes difficult to follow the way how some quantities are extracted from the simulations. In particular, while I believe that the effect of the irregularities introduced into the surface on the load-displacement curve does exist, but in my feeling the effects are not very strong. This leads to the question, in which way “randomness” in the simulations might influence the results. This concerns especially the delta parameters complied e.g. in Table 1 and used throughout the paper to quantify the effects of the different types of surface defects. I was unable to figure out how these values have been determined from the load-displacement curves. One problem seems to be the discrete increase of the displacement for the simulations and, of course, the discreteness of dislocation generation. The up and down of the load values in Figure 3 really leads to doubts whether the corresponding results are really robust against slight variations of the simulation conditions. In Table 1 the authors prose an accuracy of 0.01 N/m for the different delta values Table 1. What is the basis for this accuracy? How is P_cr really determined from the QC simulations and how sensitive are these in view of slight variations of conditions? What happens, if the stepsize for displacement is made smaller by a factor of 2?
- Thank you for your professional suggestion. We understand the doubt you mentioned that the effects may be tiny or even might be caused by the randomness. Frankly speaking, we have simulated several times to verify whether the randomness is the major factor, by which the accuracy of 0.01 N/m for the different delta values in Table 1 has been obtained. In fact, we do figure out such effects really exist. Similarly, the experimental work on indentation on gold near surface steps made by Corcoran et al showed large effects near surface steps. So it is reasonable for us to believe such effect shown in this manuscript when we indent near the surface defects.
According to the QC method, Pcr is the maximum load in the load-displacement curve which can be obtained in the output file after QC simulation. And the nanohardness (H) can be calculated by the equation H=Pmax/A [1], where Pmax(or Pcr) is the maximum load and A is the contact area when indenting.
Secondly, just as you said, the result of QC simulation can be sensitive under the various conditions such as the “tolerance” factor (which is used for solver and convergence checks), the “PROXFACT” factor (which is used to determine the range of nonlocal effects), the step size of the indentation (we choose the stepsize of 0.02nm because it facilitate to catch the elastic-to-plastic transition with the minimum steps) and so on. Therefore, we fix these factors to verify the validity of the simulation results.
[1]Nanoindentation. Available online: https://en.wikipedia.org/wiki/Nanoindentation (accessed on 2 May 2018).
4) The authors apparently do not discuss details of the orientation of the dislocation dipoles. This orientation is, for example, flipped if you compare Fig. 5 a and b vs. Fig 5c. Why is this? There is asymmetry in the system, so it cannot be accidental. The same asymmetry seems relevant also for the simulations from 3.2. In general, I have the feeling that there no symmetry induced enhancement, but asymmetry induced decrease of hardness.
- Thank you for your professional suggestion. Indeed, we believe that the orientation of the dislocation dipoles can be interesting and significant that worthy further discussion. As regards to the nanoindentation with single surface defect, we have already found out a strain competition mechanism near the load surface in order to explain the reason of perfect dislocation emission [2]. However, it would be a strongly tough work to explain the orientation of dislocation diploes that influenced by so many factors in this simulation model. It is necessary to be treated as a new topic, unfortunately not the main point of this article. Secondly, similarly as your feelings, we have been supervised and shocked at first by the simulation result that when the surface pit defects symmetrically distribute in both sides of the indenter, the nanohardness of Pt thin film gets relevantly greater value. After it is confirmed over and over again, we finally recognize the symmetry-induced enhancement shown as Fig6.
[2]Zhang, Z.L.; Ni, Y.S. Multiscale analysis of delay effect of dislocation nucleation with surface pit defect in nanoindentation. Comp. Mat. Sci. 2012, 62, 203–209.
5) I am completely lost by the discussion on page 17-18. Apparently little reference is made to section 3.1 in that later section. Two independent works put together….
- Thank you and we feel sorry for that. As it mentioned in the first comment, this work is focused on the different distribution of the surface multi-defects in nanoindentation, where the unilaterally distributed multi-defects model and the bilaterally distributed multi-defects model are indispensible complementary in order to make a comprehensively investigation. Secondly, we found two different effects respectively shows in the simulation modes, where the synergy effect in the unilaterally distributed multi-defects model (section 3.1) and the symmetry-induced enhancement effect in the bilaterally distributed multi-defects mode (section 3.2). That’s why apparently little reference is made to section 3.1 in that later section.
More detailed comments:
(a) Why did the authors study platinum? I note that platinum has a very large stacking fault energy. Is the very high value of that SFE reflected by the employed potential?
- Thanks for such professional advice. What you mentioned is totally correct that the (1 1 1) surface energy (γ111) of platinum mainly reflecting the SFE in the EAM potential, is 1.46 J/m2, which is approximately 1.7 times as large as the one of aluminum (0.869 J/m2). We choose Platinum material just because the Pt group metals and in particular platinum alloys are indispensable in many fields of industrial application because of their outstanding physical and chemical properties [3].
[3]B. J. Merker, D. Lupton, M.Topfer, H. Knake. High Temperature Mechanical Properties of the Platinum Group Metals. Platinum Metals Rev, 45(2001), 74-82
(b) Line 52: EAM means embedded atom method. Apparently there are some works by Ercolessi and Adams to determine EAM parameters, but Ref. [19] is not such a work. The Ercolessi and Adams works have appeared later. Please check!
- Thanks for your carefulness.[19] has been replaced with the following one “F. Ercolessi, J. B. Adams. Interatomic potentials from first-principles calculations: The force-matching method. Europhys. Lett. 1994, 26, 583–588”.
(c) E.g. line 62. The notion “atomic spacing” is inaccurate. The smallest distance between Pt atoms is 0.277 nm, so the “atomic spacing” cannot be smaller.
- Thanks for your advice. To be precise, the Burgers vector expresses the direction and magnitude of dislocation slip, not the smallest distance between Pt atoms. In this simulation model, perfect edge dislocation finally resolves into two Shockley partial dislocations with a more stable state after dislocation reaction [4]. Therefore, when indenting into (111) surface of Pt, the direction of the Burgers vector is , and its length is 2 times large as the atomic spacing in direction (h0 :0.1386 nm). It also can be calculated as the following formula:
Where a is the crystallographic lattice constant of Pt. It is the same for FCC metal[5,6].
[4]Zhang, Z.L.; Ni, Y.S. Multiscale analysis of delay effect of dislocation nucleation with surface pit defect in nanoindentation. Comp. Mat. Sci. 62(2012), 203–209.
[5]Zhang, Z.L.; Ni, Y.S.; Zhang, J.M.; Wang, C.; Ren, X.D. Multiscale analysis of size effect of surface pit defect in nanoindentation. Micromachines, 9(2018), 298–309.
[6]Zhang, Z.L.; Ni, Y.S.; Zhang, J.M.; Wang, C.; Jiang, K.; Ren, X.D. Multiscale simulation of surface defects influence nanoindentation by a Quasi-Continuum Method. Crystals, 8(2018), 291–300.
(d) Line 63 is not be Burgers vector but the LENGTH of the Burgers vector.
- Thanks for your carefulness. It is modified to be the length of the Burgers vector.
(e) Fig. 2 The quantities H and W are not accurately referred to/defined in the Figure caption (Figure + corresponding Figure captions should be self contained).
- Thanks for your suggestion. The quantities H and W have been defined in the Figure caption.
(f) Self contained Figure+Figure caption: A-A1 etc. are unclear from Figure 3.
- Thanks for your suggestion. Necessary explanation has been added to be clear in the caption of Figure 3.
(g) Figure 4: How is the hardness calculated? The y-axis is unlikely the “nanohardeness curve”
- According to the QC method, the nanohardness (H) can be calculated by the equation H=Pmax/A, where Pmax is the maximum load and A is the contact area when indenting. In this simulation, the maximum load in Fig3 divided by the width of the indenter can be the nanohardness, because the load in Fig.3 is already expressed as force per unit thickness (N/m), which is obtained by dividing the total force on the indenter by the repeat distance in the out-of-plane direction of the crystal structure (0.4801 nm for this model). So, the tendency of nanohardness from N0 to N3, is the same as the maximum load, result from the constant contact area.
(h) The labelling of the subfigures in Figure 5 is unclever (the same holds for Fig. 7 and 8): simply use the codes of your microstructures N1, L3R3 etc. alongside with possible points (B, B1, holds for Figure 5) directly in the subfigure or use that as the label. The can make the figure captions much more efficient and all is easier for the reader.
- Thanks for such great suggestion. It helps us a lot, and provides us a new way to make our article more efficient in the later study. The codes of our microstructures N1, L3R3 etc. have been added into the caption of Fig.5, Fig.7 and Fig.8 in order to make it easier for the readers.
(i) Line 168 and 170: there are awkward commas between the delta parameters.
- Thanks for your carefulness. It has been revised.
(j) Eq. 8-10: the unit GPa should not be italic.
- Thanks for your carefulness. It has been revised.
(k) Figure 8: In my opinion there should be a stacking fault only between the two pair Shockley dislocations constituting the dipole.
- Thanks for your carefulness. According to the punch indentation of QC method [7], when the rigid indenter driven into the (111) plane of FCC metal, the dislocation nucleate beneath the indenter tip, and the atoms nearby continuously slip along the Namely, the stacking fault gradually formed. So, it is more reasonable and visual to treat the stacking fault as a slip line from the indenter.
[7]Tadmor, E.B.; Miller, R.; Phillips, R. Nanoindentation and incipient plasticity. J. Mater. Res. 1999, 14, 2233–2250.
(l) Figure 9 is a riddle to me. What is “SFW”?.
- Thanks for your patience. SFW is short for stacking fault width.
Necessary information above has been properly added into the manuscript.
According to these comments, this manuscript has been carefully revised. Now, we respectfully resubmit our revision and hope it may be published in Materials.
Yours Sincerely,
Zhongli Zhang& Can Wang& Yushan Ni

Reviewer 2 Report
In this paper Zhongli Zhang et al. have been investigated the quasicontinuum method has been applied to probe the thin film with surface multi-defects, which is commonly seen in nanoimprint technique and bulk micromachining. Three unilaterally distributed multi-defects models and six bilaterally distributed multi-defects models of Pt thin film have been carried out in nanoindentation. The results show that the nanohardness gradually decreases as the number of unilaterally distributed multi-defects increases, also along with the increasingly low decline rate of the nanohardness. The synergy effect of the unilaterally distributed multi-defects has been highly evidenced by the critical load revision for dislocation emission of Pt thin film, and it is predicted into a universal form with the synergy coefficient among the existing multi-defects for FCC metals. Moreover, the nanohardness obviously increases when the bilaterally distributed multi-defects form into symmetrical couple, and it could be even greater than the one with defect-free surface, due to the symmetry-induced enhancement effect on nanohardness. The symmetry-induced enhancement coefficient has been brought out and has well explained the symmetry-induced enhancement effect of bilaterally distributed multi-defects on the nanohardness by a prediction formula. Further, the characteristic length of symmetric relation has been brought out to calculate the symmetry-induced enhancement coefficient and it has been effectively predicted to equal to the sum of the adjacent distance between the surface defect and the indenter, the defect depth near the indenter, and the defect width for FCC metal.
The objective of this work is of interest but the paper would benefit from some thorough editing.
General remarks: minor English spelling errors may be checked!
Some specific remarks were listed below:
If the spectroscopic analysis (ATR-FTIR or FTIR, 1H NMR, EDX, XRD …), along with simulations (theoretical analysis) can be used for further structure analysis, results will be better!!! The one of the lack of the paper are solely simulations (theoretical analysis)!!!
Generally, the structural analysis is poor, without adequate references and literature analysis, and should be updated!
The authors may also be interested to read some recent papers can provide us a lot of up-to-date information about the instrumental methods and techniques for structural characterization of different systems, as mentioned in your manuscript [Mat. Sci. Eng. C 79 (2017), 930-949].
The citing of relevant references is poor, and the one of the lack of the paper relates to the inadequate references for data interpretation (only 24)!
Taking into account the above mentioned remarks, I find it possible to recommend the article for publication in Materials, after major revisions.
Author Response
Author response
Dear Reviewer 2,
Thank you for your kind suggestions. All comments have been replied as follows.
If the spectroscopic analysis (ATR-FTIR or FTIR, 1H NMR, EDX, XRD …), along with simulations (theoretical analysis) can be used for further structure analysis, results will be better!!! The one of the lack of the paper are solely simulations (theoretical analysis)!!!Generally, the structural analysis is poor, without adequate references and literature analysis, and should be updated! The authors may also be interested to read some recent papers can provide us a lot of up-to-date information about the instrumental methods and techniques for structural characterization of different systems, as mentioned in your manuscript [Mat. Sci. Eng. C 79 (2017), 930-949]. The citing of relevant references is poor, and the one of the lack of the paper relates to the inadequate references for data interpretation (only 24)!
- Thank you for your recognition and professional suggestion about our work. We can’t agree more about what you said that it would be wonderful if the spectroscopic analysis along with simulations can be used for further structure analysis. There are two main limitations as following. Firstly, our aim of this paper is to investigate how the distribution of the surface multi-defects influences the nanohardness and its micro mechanism. Thus, it is in a great difficulty to make a Pt thin film sample with specific surface multi-defects. Any difference between those samples will influence the results. Secondly, with the deepening of theoretical basis research and the improvement of the computational material science, more and more scientists choose numerical simulation method including QC method due to the inevitable limitation of experiments, where the conditions can not be precisely and singly controlled. In this model, the substrate of thin Pt film (0.1mm thick) is rigid and the symmetry boundary conditions on the sides of the model have been applied. Perfect stick conditions between the indenter and crystal are assumed that the friction and interactions between indenter tip atoms and film atoms are not considered. The number of atoms in contact with the indenter remains constant that they cannot slip out from under the indenter. Moreover, it is necessary and significant to verify the validity of QC method. In fact, some scientific workers have already paid attention to such aspect and finished assessing its reliability a few decades ago. Shenoy et al of Brown University in USA have performed atomistic lattice statics using quasicontinuum method and meanwhile carried out the analytic analysis within the Rice-Thomson and Peierls-Nabarro frameworks, and they have found out the predications of the analytical models are qualitatively borne out by the atomistic simulations [1]. And at the next year, Knap and Ortiz have analyzed the accuracy and convergence characteristics of the quasicontinuum theory, where the effect of the summation rules on accuracy and the refinement tolerance have been assessed, and they have found that with sufficient mesh adaptivity, the quasicontinuum method is capable of simulating evolving microstructures comparable, in energetic terms, to those obtained from a full atomistic calculation [2]. Alizadeh et al have conducted two-dimensional quasicontinuum simulations of nanoindentation processes on aluminum thin films with nickel, copper, and silver coatings of different thicknesses under rectangular indenter using the embedded atom method (EAM) potentials, and the obtained results are shown to be in good agreement with limited available experimental data and the analytical Rice–Thomson and Peierls–Nabarro dislocation models [3]. Shiyun Shi et al have performed a series of experiments in order to examine the response of clamped steel plates loaded quasi-statically at their centre by a rigid rectangular indenter, where the deformation modes of the tested plates are summarized based on the experimental observations and numerical simulations [4].
Based on above discussion, the simulation results from this paper by QC method has its undeniable validity and reliability
Additionally, necessary information and some updated references have been added into the manuscript.
[1] Shenoy, V.B.; Phillips, R.; Tadmor, E.B. Nucleation of dislocations beneath a plane strain indenter. J. Mech. Phys. Solids. 2000, 48(4), 649–673.
[2] Knap, J.; Ortiz, M. An analysis of the quasicontinuum method. J. Mech. Phys. Solids. 2001, 49(9), 1899–1923.
[3] Alizadeh, O., Eshlaghi, T.O., Mohammadi, S. Nanoindentation simulation of coated aluminum thin film using quasicontinuum method, Computational Materials Science, 2016, 111, 12-22.
[4] Shi, S.Y.; Zhu, L.; Karagiozova, D.; Gao, J.Y. Experimental and numerical analysis of plates quasi-statically loaded by a rectangular indenter. Mar. Struct. 2017, 55, 62-77.
According to these comments, this manuscript has been carefully revised. Now, we respectfully resubmit our revision and hope it may be published in Materials.
Yours Sincerely,
Zhongli Zhang& Can Wang& Yushan Ni

Round 2
Reviewer 1 Report
see attached file

Author Response
Author response
Dear Reviewer 1,
Thank you for your kind suggestions. All comments have been replied as follows.
(1) Point c has not been resolved. The wording is inappropriate. Note that the length of a Burgers vector of a partial dislocation (e.g. Shockley) in fcc is NOT identical with any real interatomic spacing in the crystal structural. Note that in my opinion the notion “atomic spacing” can only be a real distance between atoms. Therefore, in my opinion use of that term in the previous paper is highly inappropriate. Please look for appropriate wording.
- Thank you for your professional advice. What you said is totally right that it is indeed inappropriate and insufficiently strict to define the notion “atomic lattice spacing”. Necessary explanation needs to be brought out. Actually, QC method is a 2D model, though the thickness of this model is 0.4801 nm, which equals to the minimal repeat distance with periodic boundary condition applied. What we want to express is exactly the adjacent distance between atoms in XY plane according to this 2D model. So, in order to precisely describe the model parameter, the sentences “the atomic lattice spacing in [1 1 1] direction (d0) is 2263 nm, the atomic spacing in direction (h0) is 0.1386 nm,” have been revised to be “when it is projected onto the XY plane, the adjacent distance between atoms in [1 1 1] direction (d0) is 0.2263 nm while in direction (h0) it is 0.1386 nm” in the manuscript. Wish it would be better. Thanks again, and we greatly appreciate you for your patience and professionality.
(2)Line 66 “=” missing prior to 66 GPa
- Thanks and “=” has been added into the text prior to 66 GPa.
(3) Figure 9 is a riddle to me. What is “SFW”?.
I have guessed that this is the stacking fault width, but the reader is not supposed to guess.
So please add this information somewhere in the figure or in the figure caption..
- Thanks and necessary explanation has been added into the figure caption.
According to these comments, this manuscript has been carefully revised. Now, we respectfully resubmit our revision and hope it may be published in Materials.
Yours Sincerely,
Zhongli Zhang& Can Wang& Yushan Ni

Reviewer 2 Report
I am not fully satisfied with the answers, but the revised version is improved!
Accept in present form!
Author Response
Author response
Dear Reviewer 2,
(1) I am not fully satisfied with the answers, but the revised version is improved!
Accept in present form!
- Thank you for your kindness. We have tried our best to revise this manuscript. A paper accepted by the SCI journal is such a tough work. We greatly appreciate you for your patience and professionality.
Yours Sincerely,
Zhongli Zhang& Can Wang& Yushan Ni
